# Running prevalence in Portugal: Socio-demographic, behavioral and psychosocial characteristics

Hugo V. Pereira[ID][1,2]*, António L. Palmeira[1,2ʘ], Eliana V. Carraça[1,2ʘ], Inês Santos[1,3ʘ], Marta M. Marques[4ʘ], Pedro J. Teixeira[1ʘ]

1 Centro Interdisciplinar para o Estudo da Performance Humana (CIPER), Faculdade de Motricidade Humana, Universidade de Lisboa, Cruz Quebrada—Dafundo, Portugal, 2 Faculdade de Educação Física e Desporto, Universidade Lusófona, Lisboa, Portugal, 3 Laboratório de Nutrição, Faculdade de Medicina, Universidade de Lisboa, Lisboa, Portugal, 4 ADAPT SFI Research Centre and Trinity Centre for Practice & Health Care Innovation, Trinity College Dublin, Dublin, Ireland

ʘ These authors contributed equally to this work.
* hugopereira@fmh.ulisboa.pt

## Abstract

The purpose of this study was to cross-sectionally estimate the prevalence of recreational running in Portugal and describe characteristics of adult recreational runners. A random representative sample of 1068 Portuguese adults was selected. Socio-demographic information, physical activity habits and running behavior were assessed. Recreational runners' training habits, motivations, barriers, vitality and flow were also assessed. The prevalence of recreational running in Portugal was 10.6%. It was higher in men (14.6% vs. 6.6%, p = .024) and in younger runners (13.6% vs. 7.7%, *p* = .026). Participants ran on average 3 times, 20 kilometers and 3 hours per week. General health orientation (88%), self-esteem (63%), and life meaning (57%) were the most predominant motives for running, while time was the most prevalent barrier (43%). This first Portuguese running prevalence representative study, indicates that almost 11% of adults ran regularly, and describes correlates of running, which can inform future running promotion interventions.

## 1. Introduction

Regular physical activity (PA) is important for staying healthy and prevent chronic diseases, such as overweight and obesity, cardiovascular disease, diabetes, and several types of cancer [1–3]. Despite the evidence about the positive relationship between PA and health, 60% of Europeans (and 74% of Portuguese) aged 15 and above, report to never or rarely exercise or play sports, and 56% (and 79% in Portugal) never or seldomly engage in PA [4]. In Portugal, objectively measured PA data indicate that 74% of Portuguese adults do not meet the World Health Organization (WHO) PA recommendations [5].

To date, considerable research has addressed the mechanisms of PA initiation among sedentary individuals, but fewer research has addressed how to support long-term maintenance and engagement among those already active [6]. PA maintenance refers to previously

**Data Availability Statement:** The dataset used for the analysis is publicly and freely available at https://osf.io/qmvws/.

**Funding:** This study was partly supported by the Fundação para a Ciência e Tecnologia, under Grant UIDB/00447/2020 to CIPER - Centro Interdisciplinar para o Estudo da Performance Humana (unit 447) and Universidade de Lisboa under PhD. grant conceded to Hugo V. Pereira. Marta M. Marques is funded by a Marie-Sklodowska-Curie Fellowship (Co-fund EDGE programme, grant agreement no. 713567). The funders had no role in study design, data collection and analysis, decision to publish, or preparation of the manuscript.

**Competing interests:** The authors have declared that no competing interests exist.

sedentary individuals who increased their PA level and maintained regular PA for at least 6 months [7]. Several contextual and individual factors influence PA maintenance. Individual factors such as motivation, goal setting, self-regulation skills (e.g. self-monitoring of behavior), and self-efficacy have been associated with sustained PA behavioral changes [8, 9]. Additionally, exercise induced flow and vitality can also contribute to exercise intrinsic reward, and may be associated with exercise adherence rates [10–12]. Finally, a favorable built environment positively influences physical activity [13]. For instance, the role of active communities, including increases in PA accessibility, routes for walking and bicycling, providing protected and suitable access to parks, sidewalks, greenways, have been shown to play a role in PA promotion [14].

Running is a unique leisure activity that requires specific behavioral self-regulation processes, which are the object of interest from the PA maintenance research community, in part because of the long hours of training and numerous running events with a large number of participants [15]. Running is one of the most popular leisure exercises [16], probably because it is inexpensive and can be performed anywhere, almost at any time. In addition, it requires little technical skills and it is easy to learn. The health benefits of running are vast, including prevention of obesity, hypertension, dyslipidemia, type 2 diabetes, osteoarthritis and hip replacement, benign prostatic hypertrophy, respiratory disease, cancer, disability, reduction of cardiovascular, and all-cause mortality [17–19].

Like other countries, Portugal has seen a steep increase in running as a PA preference, and the number of running events appears to have increased exponentially over the past 20 years. Previous data suggest a prevalence between 5.5 and 8.5% of running among the Portuguese population [20, 21]. A recent survey, concerning levels of self-reported PA and preferences of leisure-time activities of Portuguese adults, indicated that running was the preferred leisure-time PA by 18% of men and 8% of women [16]. Prevalence data from other European countries vary between approximately 30% in Denmark, 23% in Spain and England, 19% in Belgium and France, 18% in the Netherlands, 15% in Finland, 13% in Germany and 12% in Slovenia [20]. Data from USA [22] and Australia [23] suggest a 15% of participation in running and jogging activities.

Longitudinal data from a sample of runners suggest that behavioral skills, self-efficacy, social support and enjoyment may be of particular importance for the adoption of regular activity, and are likely to have a key role in encouraging running maintenance [24, 25].

In order to promote PA and running in a sustainable manner, it is crucial to understand individuals' experiences and outcomes, as well as the factors that predispose them to engage and maintain this activity. This study contributes to filling this gap in the literature by estimating the prevalence of recreational running in Portugal and describing the characteristics of adult recreational runners, including socio-demographic, behavioral (e.g., running history and patterns), and psychological (e.g. motives, barriers, vitality and flow).

## 2. Material and methods

### 2.1. Study design

This was a cross-sectional study based, through a PA and running habits survey applied during October 2017, to a Portuguese representative sample of adults.

### 2.2. Participants

Participants were selected based on a computer generated probabilistic (digit randomization) sample of telephone numbers, which were stratified by country region. The sampling unit was private residential households with a landline and/or mobile telephone. To assume

representativeness of the Portuguese population (mainland Portugal and islands) by gender and age group (18–40 yrs.; 41–65 yrs.), a sample size of 1068 individuals (267 for each gender-age group) was estimated, considering a response rate of 50%, with a 95% confidence interval and a country sampling error of 3%. Considering a previous estimate of running prevalence in the Portuguese population of about 10% [21], the expected sample of recreational runners was 106 (with a 90% confidence interval, and 5% country sampling error).

Of the 2246 initial contacts, 1150 accepted participating in the study (participation rate of 51.2%), 40 were excluded due to chronic diseases, 10 due to pregnancy, and 16 due to incomplete answers (unable to complete de questionnaire). Therefore, the final sample was constituted by 1084 eligible individuals, i.e., with Portuguese nationality and aged between 18 to 65 years. Sixty participants failed to provide valid data on weekly PA, and were not included is some PA analyses. All participants gave their informed consent before entering the study.

**2.2.1. Survey.** This study was approved by the Ethics Committee of the Faculty of Human Kinetics, University of Lisbon (CE-FMH 13/2017).

First, a panel of running and PA experts from academic and non-academic public and private institutions, agreed on a definition of recreational running. In this process, a literature-based definition [15, 26, 27] was sent to 10 experts. After a content analysis of 8 definitions, the research team arrived at the following definition: a recreational runner is someone who runs at least 2 days per week or at least 60 minutes per week, over the past 3 months, excluding any preparation for competitive sports. Then, a telephone-based survey was developed, in close collaboration with this panel. The survey assessed socio-demographic characteristics (e.g., gender, age, marital status), weekly physical activity habits (IPAQ-SF [28]) preference for non-sedentary activities (ACI [29]) attitudes toward PA (e.g., PA prevalence and health benefits), and running behavior (weekly sessions and time). For those classified as recreational runners in accordance to the definition aforementioned, running behavior (including self-reported running frequency, time, distance, location, and monitoring devices used), behavioral regulations (e.g., introjected, integrated, etc.–BREQ-3 [30]) motives for running (e.g., health or challenge–MOMS [31]), vitality levels (e.g., feelings of energy–SVS [32]), experience of flow (e.g., task focus–DFS [33]) and barriers to running (e.g., time or injuries) were also assessed. Short versions of previously validated scales, representing these different constructs were adapted for usage in epidemiological surveys through telephone interview. After eligibility checking, two questions about running frequency and volume (minutes) determined if the participant could be classified as a recreational runner. If so, running behavior, motives and regulations, as well as vitality and flow, were assessed (Full version of the questionnaire can be found in https://osf.io/qmvws/).

Data was collected by fieldwork researchers from the Institute of Environmental Health / Institute of Preventive Medicine & Public Health, Faculty of Medicine, University of Lisbon, through a 20-minute telephone interview. All researchers received equal training regarding the explanation of the goals of the study and conduction of the interviews. A quality control procedure was conducted by reapplying (by a different interviewer) the same questionnaire to 10% of the initial sample. The dataset used for the analysis is publicly and freely available at https://osf.io/qmvws/.

## 2.3. Data analysis

Statistical analyses were conducted using IBM SPSS® version 23. The significance level was set at $p < 0.05$ for all tests. Descriptive statistics were expressed in relative frequencies or mean ± standard deviation. Differences between runners and non-runners regarding socio-demographic factors, PA habits, attitudes toward PA, and differences between gender and age

groups regarding running motives, behavioral regulations, vitality, flow, and barriers for running, were analyzed using independent-sample $t$ tests and effect size (Cohen's $d$) calculations, for continuous variables and Chi-square ($\chi2$) tests, for categorical variables. Pearson correlations were used to examine associations between the motivational running behavior variables and the psychological outcomes.

## 3. Results

### 3.1. Prevalence of recreational running

The prevalence of recreational running in Portugal was 10.6%. The prevalence was higher in men when compared with women (14.6% vs. 6.6%; $\chi^2$, (1, N = 115) = 5.089, $p$ = 0.024) and in younger than in older participants (13.6% vs. 7.7%; $\chi^2$, (1, N = 115) = 4.975, $p$ = 0.026).

### 3.2. Characteristics of recreational runners

Regarding running behavior, runners reported running 3.4 ± 1.3 sessions/week, 20.0 ± 10.7 km/week, and 3.0 ± 2.3 h/week. 73% of the participants prefer to run alone (vs. 13% preferring a "running group"), 69% also do other physical activities (of which 58% reported doing warm-up and stretching exercises), 69% use technology during running sessions (of which 45% use a watch and music features), 68% report running in a natural setting (of which 58% run "on roads"). Of all runners, 21% have participated in at least one race/event (13% engaged in 2 to 5 events per year) and 15% had an injury in the previous year (with an average of a 7 weeks recovering period).

### 3.3. Age and gender comparisons

General health orientation, self-esteem, and life meaning were the most prevalent motives for running (57–88%), while (lack of) time was the most prevalent barrier (43%). When comparing motives and behavioral regulations across gender and age groups (Table 1), the only significant differences detected were that younger people tend to run more for the feeling of competition ($p$ = 0.01) and personal goal achievement ($p$ = 0.03) motives.

**Table 1. Motives and behavioral regulations across age and gender groups of runners.**

|  |  | Female (36) | Male (79) |  |  |  | 18–40 Yrs. (73) | 41–65 Yrs. (42) |  |  |  |
|---|---|---|---|---|---|---|---|---|---|---|---|
|  |  | M(SD) | M(SD) | T | P | d | M(SD) | M(SD) | T | P | d |
| MOMS | Psychological coping | 2.6 (0.9) | 2.6 (0.8) | -0.40 | .69 | .08 | 2.7 (0.8) | 2.4 (0.9) | 1.92 | .06 | .36 |
|  | Self-esteem | 3.4 (1.1) | 3.4 (1.0) | 0.36 | .72 | .07 | 3.5 (1.0) | 3.2 (1.1) | 1.28 | .20 | .24 |
|  | Life meaning | 3.4 (0.9) | 3.4 (0.9) | -0.07 | .95 | .03 | 3.4 (0.8) | 3.3 (0.9) | 0.77 | .44 | .15 |
|  | General health orientation | 4.3 (0.8) | 4.1 (0.7) | 1.32 | .19 | .25 | 4.2 (0.7) | 4.2 (0.8) | -0.07 | .94 | .01 |
|  | Weight concern | 2.2 (1.1) | 2.3 (1.0) | -0.32 | .75 | .06 | 2.2 (1.1) | 2.2 (0.9) | 0.01 | .99 | .00 |
|  | Recognition | 2.2 (0.8) | 2.0 (0.8) | 1.65 | .10 | .31 | 2.0 (0.9) | 2.0 (0.7) | -0.09 | .93 | .02 |
|  | Competition | 2.2 (0.9) | 2.4 (1.0) | -0.89 | .38 | .17 | 2.5 (1.0) | 2.0 (0.8) | 2.83 | .01 | .53 |
|  | Affiliation | 2.2 (1.1) | 2.3 (1.0) | -0.32 | .75 | .06 | 2.2 (1.1) | 2.2 (0.9) | 0.01 | .99 | .00 |
|  | Personal goal achievement | 2.1 (0.9) | 2.3 (1.0) | -1.29 | .20 | .24 | 2.4 (1.1) | 2.0 (0.8) | 2.23 | .03 | .42 |
| BREQ-3 | External regulation | 1.6 (0.8) | 1.8 (0.7) | -1.09 | .28 | .21 | 1.7 (0.8) | 1.8 (0.6) | -0.61 | .54 | .12 |
|  | Introjected regulation | 2.7 (0.9) | 2.6 (0.9) | 0.61 | .55 | .11 | 2.7 (1.0) | 2.5 (0.7) | 1.60 | .11 | .30 |
|  | Identified regulation | 4.3 (0.8) | 4.2 (0.5) | 0.71 | .48 | .13 | 4.2 (0.6) | 4.1 (0.6) | 1.01 | .31 | .19 |
|  | Integrated regulation | 3.6 (1.0) | 3.5 (1.0) | 0.53 | .60 | .10 | 3.4 (1.0) | 3.7 (1.0) | -1.39 | .17 | .26 |
|  | Intrinsic motivation | 4.1 (0.8) | 3.8 (0.8) | 1.72 | .09 | .32 | 3.9 (0.8) | 4.0 (0.7) | -0.23 | .82 | .04 |

**Table 2. Correlation between motives and behavioral regulations with quantitative measures of running behavior.**

|  |  | Weekly frequency | Weekly distance (km) | Weekly time (min) |
|---|---|---|---|---|
| MOMS | Psychological coping | -0.13 | -0.08 | 0.03 |
|  | Self-esteem | -0.09 | -0.14 | -0.02 |
|  | Life meaning | -0.01 | 0.14 | -0.08 |
|  | General health orientation | -0.03 | 0.03 | -0.06 |
|  | Weight concern | 0.00 | 0.16 | -0.15 |
|  | Recognition | 0.09 | 0.07 | -0.10 |
|  | Competition | 0.07 | 0.15 | 0.04 |
|  | Affiliation | 0.00 | 0.16 | -0.15 |
|  | Personal goal achievement | 0.13 | 0.25* | 0.02 |
| BREQ-3 | External regulation | 0.01 | -0.12 | -0.07 |
|  | Introjected regulation | 0.06 | -0.05 | -0.09 |
|  | Identified regulation | 0.00 | 0.09 | 0.05 |
|  | Integrated regulation | 0.16 | 0.13 | -0.08 |
|  | Intrinsic motivation | 0.03 | 0.03 | -0.06 |

N = 115

* p < .05.

**3.3.1. Running motivation, behavior and psychological outcomes.** Table 2 presents associations of motives and behavioral regulations with running behavior, namely weekly running frequency, distance and time. Personal goal achievement was associated with weekly distance (r (113) = .253; $p < .014$) but no other significant association was noted.

Associations of motives and behavioral regulations with vitality and experience of flow are shown in Table 3. More autonomous forms of motivation for running (identified, integrated, and intrinsic), and "Life Meaning" and "General Health Orientation" motives were associated with higher vitality and experience of flow.

**Table 3. Correlations between motives and regulations with vitality and flow.**

|  |  | Vitality | Flow |
|---|---|---|---|
| MOMS | Psychological coping | 0.03 | 0.05 |
|  | Self-esteem | 0.08 | 0.18 |
|  | Life meaning | 0.24* | 0.28** |
|  | General health orientation | 0.28** | 0.42** |
|  | Weight concern | 0.13 | 0.12 |
|  | Recognition | 0.18 | 0.13 |
|  | Competition | 0.01 | 0.03 |
|  | Affiliation | 0.13 | 0.12 |
|  | Personal goal achievement | -0.04 | 0.06 |
| BREQ-3 | External regulation | 0.05 | 0.04 |
|  | Introjected regulation | 0.11 | 0.16 |
|  | Identified regulation | 0.36** | 0.47** |
|  | Integrated regulation | 0.53** | 0.53** |
|  | Intrinsic motivation | 0.43** | 0.45** |

N = 115

** P < .001

* p < .05.

**Table 4. Weekly PA levels in recreational runners vs. non-runners in minutes/week.**

|  | Non-runners (910) | Runners (114) | T | P | d |
|---|---|---|---|---|---|
| Vigorous PA (min/wk) | 32.3 (52.5) | 72.3 (46.6) | -7.77 | < .001 | .49 |
| Moderate PA (min/wk) | 79.3 (68.7) | 73.0 (67.1) | 0.93 | .35 | .06 |
| Walking (min/wk) | 47.7 (51.7) | 51.9 (58.9) | -1.00 | .32 | .06 |
| Activity Choice Index | 2.6 (0.7) | 2.9 (0.7) | -3.00 | < .003 | .19 |

### 3.4. Runners vs. non-runners

The percentage of runners in a civil partnership/marriage was lower than in the non-runner sample (49.0% vs. 66.0%; $\chi^2$, (1, N = 917) = 11.38, $p$ = .001). There was a higher percentage of runners with a third level education degree (64.63% vs. 41.38%; $\chi^2$, (1, N = 790) = 16.14, $p <$ .001) and earning more than 1456€ per month (46.9% vs. 33.3%; $\chi^2$, (1, N = 1025) = 8.06, $p$ = .005) in comparison to non-runners.

Table 4 shows weekly PA levels of recreational runners vs. non-runners. Sixty participants failed to provide valid data on weekly PA. Although vigorous PA, measured in minutes, was higher in runners ($p < 0.001$), there were no differences between runners and non-runners regarding moderate PA or walking. Regarding the preference for non-sedentary activities, runners presented a higher overall ACI score than non-runners ($p < 0.05$). About 75% of runners reported selecting the stairs instead of the elevator, stand instead of seating, and to walk instead of driving in their daily routine.

Table 5 presents attitudes toward PA of recreational runners and non-runners. Almost all Portuguese individuals believe that PA increases quality of life but, surprisingly, only 3% knew the current WHO PA recommendations for adults. The great majority of runners enjoy doing sports and PA and believe that more people are nowadays engaging in PA, and between 80–90% believe that active commuting is important and state having a group friends to do PA with. Compared with non-runners, runners reported enjoying PA more frequently, recognizing that now, there are more people engaging in PA, having friends who can do PA with, and more free PA opportunities (all $p < 0.05$).

Regarding health perception, 82% percent of runners state having a good or excellent health, comparing to 58% of non-runners ($\chi^2$, (1, N = 1021) = 23.40, $p < .001$).

## 4. Discussion

The present study revealed that in Portugal 10.6% (14.6%—men; 6.6%—women) of adult individuals were recreational runners by 2017. The prevalence was higher in men compared to women and in younger participants compared to the older population group. Runners reported running on average approximately 3 times, 20 kilometers and 3 hours per week. The

**Table 5. PA attitudes of recreational runners vs. non-runners.**

|  | non-Runners (910) | Runners (114) | Comparison |
|---|---|---|---|
| PA increases quality of life | 99% | 99% | $\chi^2$, (2, N = 1035) = 1.05, $P$ = .59 |
| Enjoy doing sport and PA | 87% | 98% | $\chi^2$, (2, N = 1035) = 22.66, $P < .001$ |
| More people are now engaging in PA | 92% | 95% | $\chi^2$, (2, N = 1035) = 8.51, $P < .05$ |
| Active commuting is important | 93% | 89% | $\chi^2$, (2, N = 1035) = 5.96, $P$ = .051 |
| Having a group of PA friends | 69% | 84% | $\chi^2$, (2, N = 1035) = 15.82, $P < .001$ |
| There are free PA opportunities | 58% | 74% | $\chi^2$, (2, N = 1035) = 11.06, $P < .05$ |
| Inability to do PA at the moment | 18% | 4% | $\chi^2$, (2, N = 1035) = 21.94, $P < .001$ |

prevalence of running behavior among Portuguese adults found in this study, is slightly lower than the preferences for running in the leisure-time activities survey previously published [16]. Moreover, it is lower than those found in other countries, which range from approximately 30% in Denmark and 12% in Slovenia [20]. Data from USA [22] and Australia [23] suggest a 15% of participation in running/ jogging activities. Due to differences in definitions and survey methodology, levels of participation in running cannot be rigorously comparable.

This prevalence result is higher than the one reported by Scheerder and coworkers [20] who previously estimated running prevalence in Portugal, by multiplying the running percentages in Spain with their sport-participation rate, based on Eurobarometer data, and then multiplying it by Portugal's sport-participation rate. According to this estimation, the running participation rate for Portugal in 2015 was then 5.5%. Results are also higher than those emerging from another data set, in which the prevalence was 8.5% [21]. However, evaluation approaches are not entirely comparable. Gender differences in running participation are similar to those found previously [21] and reflect gender inequalities in overall PA involvement [4].

Relative to the running motives, results suggest that intrinsic motives (general health orientation, self-esteem and life meaning) are more prevalent than controlled ones (weight concern, recognition and personal goal achievement) in recreational runners. Our data also suggests that this set of runners presents higher score in autonomous forms of behavior regulation (intrinsic, integrated and identified). Previous research about running motives suggests physical and mental health as the main motives for engaging in running [34–37]. Additionally, these findings are similar with those found with female ultra-runners, proposing general health orientation, self-esteem and psychological coping as the strongest motivational factors [35], with half, full and ultra-marathoners, identifying health orientation, personal goal achievement and self-esteem [36]. Besides health and wellbeing, one other study suggests challenge [38] as the main motive for engaging in running but these results were not confirmed by our data.

The difference between younger and older runners' motives (younger being more motivated by competition and personal goal achievement) was previously analyzed by Masters and colleagues [38], who indicated that first time ("rookie") runners were more concerned with health, weight and personal goal achievement. More recently, it was found that younger runners were more motivated by personal goal achievement, such as running to beat personal best times [15].

Our data suggests that the most prevalent barrier is lack of time. Although we did not find any study about barriers towards running exercise in already active individuals, lack of time was reported as the main obstacle to physical activity among inactive adults [39, 40].

Besides an association between personal goal achievement motive and weekly distance ran by the participants, no other relation between motivations and running behavior measures (weekly running distance and time, and years of running experience) was found. Although there was low association between motivational variables and quantitative measures of running behavior, intrinsic motives, together with autonomous forms of behavior regulation were related to positive health outcomes (vitality and flow). Similar results were found longitudinally in a sample of runners [41], in systematic reviews [42, 43], and agree with self-determination theory (SDT) basic tenets [44], by which qualitative aspects of motivation should have a closer association with qualitative aspects of the behavior, rather than with its amount (quantitative aspect).

This data suggests recreational runners report higher VPA compared to non-runners, but not MPA and walking. Walking and MPA comparison might be hindered by IPAQ-SF overestimation, and small accuracy of this instrument at moderate PA levels [28]. There are some differences in reported running and IPAQ-SF measures for VPA, probably due to overestimation

of running or methodologic issues related to IPAQ-SF calculations [28]. Regarding attitudes towards PA, it seems that both runners and non-runners believe in PA health-enhancing potential, but runners tend to enjoy PA, have people with whom to practice and recognize PA opportunities more frequently than non-runners.

Although our results confirm previous studies on physical activity motivation, the use of self-reported instruments to estimate running and weekly PA might lead to some bias, and the cross-sectional nature of this investigation prevents determining the causal direction of the associations. Longer prospective and intervention studies are thus required to clarify how motivational dynamics influence psychological wellbeing and also running maintenance.

## 5. Conclusions

This is the first Portuguese running prevalence study with a representative sample and the first to address psychological determinants of recreational running. The prevalence can be considered high (about one in every 10 adults runs regularly for exercise) and may have increased since 2017. Moreover, results suggest that intrinsic motives (general health orientation, self-esteem, and life meaning) and autonomous forms of behavior regulation (intrinsic, integrated, and identified) are significant for these runners. With this in mind, public policies and marketing efforts could target these constructs, aiming to promote recreational running initiation and/or maintenance, by helping runners to find their own motivation, through the satisfaction of the three psychological needs (competence, autonomy, and relatedness) identified by SDT, and training self-regulation strategies.

## Acknowledgments

The authors acknowledge Violeta Alarcão and the Institute of Environmental Health / Institute of Preventive Medicine & Public Health, Faculty of Medicine, University of Lisbon, for the collaboration in data collection.

## Author Contributions

**Conceptualization:** Hugo V. Pereira, António L. Palmeira, Eliana V. Carraça, Inês Santos, Marta M. Marques, Pedro J. Teixeira.

**Data curation:** Hugo V. Pereira.

**Formal analysis:** Hugo V. Pereira, António L. Palmeira.

**Funding acquisition:** Pedro J. Teixeira.

**Investigation:** Hugo V. Pereira, António L. Palmeira, Inês Santos, Marta M. Marques, Pedro J. Teixeira.

**Methodology:** Hugo V. Pereira, António L. Palmeira, Eliana V. Carraça, Inês Santos, Marta M. Marques, Pedro J. Teixeira.

**Project administration:** Hugo V. Pereira, António L. Palmeira, Pedro J. Teixeira.

**Resources:** Hugo V. Pereira, António L. Palmeira, Pedro J. Teixeira.

**Supervision:** António L. Palmeira, Marta M. Marques, Pedro J. Teixeira.

**Validation:** António L. Palmeira, Pedro J. Teixeira.

**Writing – original draft:** Hugo V. Pereira, António L. Palmeira.

**Writing – review & editing:** Hugo V. Pereira, António L. Palmeira, Eliana V. Carraça, Inês Santos, Marta M. Marques, Pedro J. Teixeira.

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
