## [Decision Letter · Decision Letter 0]

29 Oct 2020

PONE-D-20-29049

Running prevalence in Portugal: Demographic, behavioral and psychosocial characteristics

PLOS ONE

Dear Dr. Pereira,

Thank you for submitting your manuscript to PLOS ONE. After careful consideration, we feel that it has merit but does not fully meet PLOS ONE’s publication criteria as it currently stands. Therefore, we invite you to submit a revised version of the manuscript that addresses the points raised during the review process.

Please, revise your methodology to truly meet the standards of epidemiological studies.

We look forward to receiving your revised manuscript.

Kind regards,

Daniel Boullosa

Academic Editor

PLOS ONE

Journal Requirements:

2.We note that you have indicated that data from this study are available upon request. PLOS only allows data to be available upon request if there are legal or ethical restrictions on sharing data publicly. For more information on unacceptable data access restrictions, please see http://journals.plos.org/plosone/s/data-availability#loc-unacceptable-data-access-restrictions.

3.Thank you for stating the following in the Acknowledgments Section of your manuscript:

[This study was partly supported by the Fundação para a Ciência e Tecnologia, under Grant

UIDB/00447/2020 to CIPER - Centro Interdisciplinar para o Estudo da Performance Humana

(unit 447) and Universidade de Lisboa under PhD. grant conceded to Hugo V. Pereira.

Marta M. Marques is funded by a Marie-Sklodowska-Curie Fellowship (Co-fund EDGE

programme, grant agreement no. 713567).]

 [The funders had no role in study design, data collection and analysis, decision to publish, or preparation of the manuscript.]

Reviewers' comments:

Reviewer's Responses to Questions

**Comments to the Author**

1. Is the manuscript technically sound, and do the data support the conclusions?

Reviewer #1: Yes

Reviewer #2: Yes

2. Has the statistical analysis been performed appropriately and rigorously? 

Reviewer #1: Yes

Reviewer #2: Yes

3. Have the authors made all data underlying the findings in their manuscript fully available?

Reviewer #1: Yes

Reviewer #2: Yes

4. Is the manuscript presented in an intelligible fashion and written in standard English?

Reviewer #1: Yes

Reviewer #2: Yes

5. Review Comments to the Author

Reviewer #1: First of all, I would like to congratulate the authors for the manuscript submitted. All the comments and suggestions I provide below have the only purpose of increasing the quality of this work and the future research lines from a constructive rationale.

Physical inactivity is considered the fourth leading cause of death worldwide, causing 5-10% of total mortality in Europe, depending on countries. Despite the ACSM, the AHA and the WHO strongly tried to reduce physical inactivity levels and sedentary behaviour, the majority of the people who live in industrialized societies do not meet the established recommendations for an active and healthy lifestyle. To better manage this situation, it is important to know not only individual factors, but also the social and economic background which leads to reduced levels of physical inactivity. In this sense, I suggest authors to include along the introduction section the concept of “active communities”, since a favourable environment (e.g. parks, running/walking paths, bike lanes or other sport facilities) seems to play an important role in the promotion of higher physical activity levels.

Regarding material and methods section, the final sample was composed by 1084 eligible individuals (after exclusions). However, Table 4 shows data of 1024 participants (910 non-runners vs 114 runners). Further, Table 1 presents data of runners sort by gender and age group. However, the runners’ group is constituted by 115 participants in this case (79 male and 36 female; 73 between 18-40 y and 42 between 41-65 y) instead of 114, like it does in Table 4. Tables 2, 3 and 5 do not display the sample size for any group. Moreover, the main result of the present investigation (i.e. the prevalence of recreational running in Portugal) is calculated over the 1084 eligible individuals, since 10.6% of this sample size results in 114 runners. The authors should clarify these issues and better explain the procedure, including specific exclusion and inclusion criteria for a better reader’s comprehension.

In the results section, I suggest authors to include the units of measurement (e.g. km, min, etc.) in the Tables, where applicable, for a better comprehension.

With regard to discussion section, at the end of the second paragraph, it is said that previous research estimates running prevalence in Portugal about 5.5% in 2015. Data derived from the present study showed that running prevalence was almost doubled within only 2 years (i.e. up to 10.6% in 2017). What do authors think about this issue, since the increase in the participation in running events already exponentially increased worldwide in the past two decades? Is the magnificent increment in the present study maybe not be comparable to previously reported data in Portugal due to different ways of assessment?

Authors should change the name of section 5 to “conclusions” instead of “discussion” (discussion is already presented in section 4). I would also suggest authors to include in the conclusions section the limitations of the present investigation.

Authors are also encouraged to revise PLOS ONE submission guidelines.

Reviewer #2: General comments:

Thank you for the opportunity to review the submitted paper “Running prevalence in Portugal: Demographic, behavioral and psychosocial characteristics”. The purpose of this study was estimate the prevalence of recreational running in Portugal and describe characteristics of recreational runners. The main results indicate that a portion of 10.6% of the population of Portugal practices running regularly, describing their main motivations and also important socio-demographic data for public knowledge. In view of the importance of disseminating descriptive data on the population, some observations should be taken into account for future publication. In the methods section, it is important to describe in more detail how the questions were structured, perhaps inserting supplementary material so that the research can be replicated. The results of the study are interesting. I imagine that they can be used in the construction of investment strategies in public policies and business marketing strategies; however, this discussion could be further explored in the text. The results could include an analysis by region in Portugal; different local characteristics can increase the information of the study. The conclusion of the study must be restructured. This chapter should be more direct. I suggest bringing information about the main motivation and something more related to the main results of the study.

Major Issues

In Title - The term "Socio-demographic" is always used in the text and that is different from the term used in the title (“Demographic”). I suggest standardize the term used.

In P.5 -Where did the participants' phone numbers come from? How was randomization performed?

In P.5 Can the data for estimating the prevalence of running from the Portuguese Cardiology Foundation be found on any website, magazine or elsewhere?

In P.5 Did all contacted patients accept to participate in the study? From what is described it seems that yes, but that those who did not answer completely did not accept to participate in the study or had any difficulty in answering the questions? It is important to describe this flow of participants in detail; I suggest a figure to help explain it.

In P.5- “…First, a panel of running and PA experts from academic and non-academic public and private institutions, agreed on a definition of recreational running – “running at least 2 days per week or at least 60 minutes per week, over the past 3 months, excluding any preparation for competitive sports…”. – This is important data from this study and cannot be defined in a generic way, a reference is needed to better support this classification.

In P.9 - “The prevalence was higher in men compared to women” - Could you move forward in this discussion? Is the relative female population of Portugal proportional to the result found or is it really larger? What is the possible explanation for the result?

In Table 3. - The methods describe "The significance level was set at p <0.05 for all tests." Why is 0.001 used in the Table 3?

Minor issues

In P. 7 - Review “10,6%”- 10.6%

In P. 9 – “eighty-two”- I suggest changing it to 82%.

In P.11 - Check the chapter 5 title.

In general – Review the reference format.

6. PLOS authors have the option to publish the peer review history of their article (what does this mean?). If published, this will include your full peer review and any attached files.

Reviewer #1: No

Reviewer #2: **Yes: **Rodrigo Gomes da Rosa

---

## [Author Response · Author response to Decision Letter 0]

8 Dec 2020

Running prevalence in Portugal: Demographic, behavioural and psychosocial characteristics

Response to the Editor and Reviewers

Editor

Answer: We have renamed all files according to PLOS ONE’s style template and verified the other style requirements.

We note that you have indicated that data from this study are available upon request. PLOS only allows data to be available upon request if there are legal or ethical restrictions on sharing data publicly.

Answer: We thank the editor for the clarification. As requested, we have uploaded an anonymised dataset at https://osf.io/qmvws/, and added this information to the manuscript as follows “The dataset used for the analysis is publicly and freely available at https://osf.io/qmvws/”.

[The funders had no role in study design, data collection and analysis, decision to publish, or preparation of the manuscript.]

Answer: We removed all funding-related text from the manuscript and included our funding statements in our cover letter.

We would like to thank the reviewers for their comments and suggestions, and for taking the time to revise this manuscript under the exceptional circumstances we are facing. All suggestions have been considered and addressed and we believe this has resulted in a substantially improved manuscript.

Reviewer 1

Physical inactivity is considered the fourth leading cause of death worldwide, causing 5-10% of total mortality in Europe, depending on countries. Despite the ACSM, the AHA and the WHO strongly tried to reduce physical inactivity levels and sedentary behaviour, the majority of the people who live in industrialized societies do not meet the established recommendations for an active and healthy lifestyle. To better manage this situation, it is important to know not only individual factors, but also the social and economic background which leads to reduced levels of physical inactivity. In this sense, I suggest authors to include along the introduction section the concept of “active communities”, since a favourable environment (e.g. parks, running/walking paths, bike lanes or other sport facilities) seems to play an important role in the promotion of higher physical activity levels.

Answer: We appreciate the insightful information. Our study aims to expand the understanding of behavioural, socio-demographic and motivational characteristics of recreational runners. Although it goes beyond the scope of our study, we considered that the concept of active communities and environmental determinants of PA is worth mentioning and have now included it in the introduction section (page 2): “Finally, a favorable built environment positively influences physical activity.(13) For instance, the role of active communities, including increases in PA accessibility, routes for walking and bicycling, providing protected and suitable access to parks, sidewalks, greenways, have been shown to play a role in PA promotion.(14)”

Regarding material and methods section, the final sample was composed by 1084 eligible individuals (after exclusions). However, Table 4 shows data of 1024 participants (910 non-runners vs 114 runners). Further, Table 1 presents data of runners sort by gender and age group. However, the runners’ group is constituted by 115 participants in this case (79 male and 36 female; 73 between 18-40 y and 42 between 41-65 y) instead of 114, like it does in Table 4. Tables 2, 3 and 5 do not display the sample size for any group. Moreover, the main result of the present investigation (i.e. the prevalence of recreational running in Portugal) is calculated over the 1084 eligible individuals, since 10.6% of this sample size results in 114 runners. The authors should clarify these issues and better explain the procedure, including specific exclusion and inclusion criteria for a better reader’s comprehension.

Answer: Thank you for the comment. The text wasn’t completely clear in this regard and this is an important point. Of the 1084 eligible participants, 115 were classified (based on the two initial questions about running behavior) as “recreational runners” (10.6%). Unfortunately, not all of the eligible participants had valid PA data. As we now describe in the material and methods section, 60 participants failed to provide valid data on weekly PA, leaving a sample of 1024. Therefore, Table 4 presents IPAQ-SF and ACI data for the 910 non-runners and 114 runners that had valid data. Additionally, we have now inserted the sample size for tables 2, 3 and 5. We have also clarified this in the Methods section (page 4): “Of the 2246 initial contacts, 1150 accepted participating in the study (participation rate of 51.2%), 40 were excluded due to chronic diseases, 10 due to pregnancy, and 16 due to incomplete answers (unable to complete de questionnaire). Therefore, the final sample was constituted by 1084 eligible individuals, i.e., with Portuguese nationality and aged between 18 to 65 years. Sixty participants failed to provide valid data on weekly PA and were not included is some PA analyses. All participants gave their informed consent before entering the study.”

In the results section, I suggest authors to include the units of measurement (e.g. km, min, etc.) in the Tables, where applicable, for a better comprehension.

Answer: As suggested, we have now included the measurement units in the tables.

With regard to discussion section, at the end of the second paragraph, it is said that previous research estimates running prevalence in Portugal about 5.5% in 2015. Data derived from the present study showed that running prevalence was almost doubled within only 2 years (i.e. up to 10.6% in 2017). What do authors think about this issue, since the increase in the participation in running events already exponentially increased worldwide in the past two decades? Is the magnificent increment in the present study maybe not be comparable to previously reported data in Portugal due to different ways of assessment?

Answer: Thank you. In fact, the methodology of the estimations was quite different. Scheerder and coworkers (20) estimated running prevalence in Portugal by multiplying the running percentages in Spain with their sport-participation rate, based on Eurobarometer data, and then multiplying it by Portugal’s sport-participation rate. In the second study, a different definition was used (21). Our definition of recreational running was not used in the previous study. We have explained in the discussion that, although running prevalence seems to have increased since the last estimation, evaluation approaches are not entirely comparable (page 9): “According to this estimation, the running participation rate for Portugal in 2015 was then 5.5%. Results are also higher than those emerging from another data set, in which the prevalence was 8.5%. (21) However, evaluation approaches are not entirely comparable. Gender differences in running participation are similar to those found previously (21) and reflect gender inequalities in overall PA involvement. (4)”

Authors should change the name of section 5 to “conclusions” instead of “discussion” (discussion is already presented in section 4).

Answer: We thank the reviewer for pointing this out. We have now changed the name of section 5.

I would also suggest authors to include in the conclusions section the limitations of the present investigation.

Answer: We have now added a paragraph with limitations of the study in the Discussion section (page 11) – “Although our results confirm previous studies on PA motivation, the use of self-reported instruments to estimate running and weekly PA might lead to some bias and the cross-sectional nature of this investigation prevents determining the causal direction of the associations. Longer prospective and intervention studies are thus required to clarify how motivational dynamics influence psychological wellbeing and also running maintenance.”

Reviewer 2

In the methods section, it is important to describe in more detail how the questions were structured, perhaps inserting supplementary material so that the research can be replicated.

Answer: Thank you for this suggestion. We have described the questionnaire structure in more detail in the Methods section (page 5), and now we have uploaded the full version of the questionnaire to https://osf.io/qmvws/. – “After eligibility checking, two questions about running frequency and volume (minutes) determined if the participant could be classified as a recreational runner. If so, running behavior, motives and regulations, as well as vitality and flow, were assessed (Full version of the questionnaire can be found in https://osf.io/qmvws/).”

The results of the study are interesting. I imagine that they can be used in the construction of investment strategies in public policies and business marketing strategies; however, this discussion could be further explored in the text.

Answer: We thank the reviewer for this suggestion. We have linked the short summary of our results with a recommendation for public policies and marketing efforts aiming at running behaviour promotion – “Moreover, results suggest that intrinsic motives (general health orientation, self-esteem, and life meaning) and autonomous forms of behaviour regulation (intrinsic, integrated, and identified) are significant for these runners. With this in mind, public policies and marketing efforts could target these constructs, aiming to promote recreational running initiation and/or maintenance, by helping runners to find their own motivation through the satisfaction of the three psychological needs (competence, autonomy, and relatedness) identified by SDT, and training self-regulation strategies.”

The results could include an analysis by region in Portugal; different local characteristics can increase the information of the study.

Answer: This is a valuable suggestion. Representativeness of the Portuguese population (mainland Portugal and islands) by gender and age group (18-40 yrs.; 41-65 yrs.) was assured but, unfortunately, the collected data does not allow region differentiations. Although data was stratified by region, some regions presented very limited number of runners, limiting regional comparisons. We intend to consider this in a future study.

The conclusion of the study must be restructured. This chapter should be more direct. I suggest bringing information about the main motivation and something more related to the main results of the study.

Answer: Thank you for this recommendation. As mentioned above, we have adjusted the conclusion and also highlighted our findings regarding runners’ motivation, linking them with the suggestions for future marketing strategies.

In Title - The term "Socio-demographic" is always used in the text and that is different from the term used in the title (“Demographic”). I suggest standardize the term used.

Answer: Thank you for noticing this. We have corrected the title, choosing “socio-demographic”.

In P.5 -Where did the participants' phone numbers come from? How was randomization performed?

Answer: Thank you. To clarify this, we have now added the following information to the methods section (page 4): “Participants were selected based on a computer generated probabilistic (digit randomization) sample of telephone numbers, which were stratified by country region.”

In P.5 Can the data for estimating the prevalence of running from the Portuguese Cardiology Foundation be found on any website, magazine or elsewhere?

Answer: We thank the reviewer for pointing this. The running prevalence data from the Portuguese Cardiology Foundation was analysed in a Master’s Thesis (Santos M. Prevalência e características sociodemográficas dos praticantes de corrida em Portugal.: Universidade de Lisboa - Faculdade de Motricidade Humana; 2017), and results are accessible here (https://www.repository.utl.pt/handle/10400.5/14297). Considering data presented in this thesis, we decided to update both the introduction (page 3) and discussion (page 9).

In P.5 Did all contacted patients accept to participate in the study? From what is described it seems that yes, but that those who did not answer completely did not accept to participate in the study or had any difficulty in answering the questions? It is important to describe this flow of participants in detail; I suggest a figure to help explain it.

Answer: Thank you for your suggestion. We didn’t report the participation rate, but we have now added this information to the Methods section (page 4) – “Of the 2246 initial contacts, 1150 accepted participating in the study (participation rate of 51.2%), 40 were excluded due to chronic diseases, 10 due to pregnancy, and 16 due to incomplete answers (unable to complete de questionnaire). Therefore, the final sample was constituted by 1084 eligible individuals, i.e., with Portuguese nationality and aged between 18 to 65 years.”

In P.5- “…First, a panel of running and PA experts from academic and non-academic public and private institutions, agreed on a definition of recreational running – “running at least 2 days per week or at least 60 minutes per week, over the past 3 months, excluding any preparation for competitive sports…”. – This is important data from this study and cannot be defined in a generic way, a reference is needed to better support this classification.

Answer: We couldn’t find a reference to support a “recreational runner” criterion when we implemented this survey. Most studies we consulted left this decision to the participant, or used samples that were participating in a public running event. We were not happy with these criteria, as we also wanted to reach participants that run recreationally without participating in public events, and we wanted to have a clear criterion in our study. Hence, we reached specialists to produce a consensus for this and future studies. We further explained the expert consultation process and mentioned the references upon which the first definition was built. Hope it further clarified the process (pages 4-5) – “First, a panel of running and PA experts from academic and non-academic public and private institutions, agreed on a definition of recreational running. In this process, a literature-based definition (26-28) was sent to 10 experts. After a content analysis of 8 definitions, the research team arrived at the following definition: a recreational runner is someone who runs at least 2 days per week or at least 60 minutes per week, over the past 3 months, excluding any preparation for competitive sports.”

In P.9 - “The prevalence was higher in men compared to women” - Could you move forward in this discussion? Is the relative female population of Portugal proportional to the result found or is it really larger? What is the possible explanation for the result?

Answer: According to the last Portuguese Census (2011), 52.2% of inhabitants were women. Nonetheless, the gender representativeness of the survey took this into account. Additionally, gender differences are similar to those found previously both among runners and in general PA. We have now added this in the Discussion section (page 9): “Gender differences in running participation are similar to those found previously (19) and reflect gender inequalities in overall PA involvement (4).”

In Table 3. - The methods describe "The significance level was set at p <0.05 for all tests." Why is 0.001 used in the Table 3?

Answer: Thank you for this observation. The minimum significance level for all tests was indeed 0.05, but in table 3 we observed a few significant results at the 0.001level, so we decided to include that information in this table. This is a common practice, suggested by institutions such as the APA.

In P. 7 - Review “10,6%”- 10.6%

In P. 9 – “eighty-two”- I suggest changing it to 82%.

In P.11 - Check the chapter 5 title.

In general – Review the reference format.

Answer: Thank you for these suggestions, we have changed the text accordingly.

---

## [Decision Letter · Decision Letter 1]

26 Dec 2020

Running prevalence in Portugal: Socio-demographic, behavioral and psychosocial characteristics

PONE-D-20-29049R1

Dear Dr. Pereira,

We’re pleased to inform you that your manuscript has been judged scientifically suitable for publication and will be formally accepted for publication once it meets all outstanding technical requirements.

Kind regards,

Daniel Boullosa

Academic Editor

PLOS ONE

Additional Editor Comments (optional):

Reviewers' comments:

Reviewer's Responses to Questions

**Comments to the Author**

1. If the authors have adequately addressed your comments raised in a previous round of review and you feel that this manuscript is now acceptable for publication, you may indicate that here to bypass the “Comments to the Author” section, enter your conflict of interest statement in the “Confidential to Editor” section, and submit your "Accept" recommendation.

Reviewer #1: All comments have been addressed

Reviewer #2: All comments have been addressed

2. Is the manuscript technically sound, and do the data support the conclusions?

Reviewer #1: (No Response)

Reviewer #2: Yes

3. Has the statistical analysis been performed appropriately and rigorously? 

Reviewer #1: (No Response)

Reviewer #2: Yes

4. Have the authors made all data underlying the findings in their manuscript fully available?

Reviewer #1: (No Response)

Reviewer #2: Yes

5. Is the manuscript presented in an intelligible fashion and written in standard English?

Reviewer #1: (No Response)

Reviewer #2: Yes

6. Review Comments to the Author

Reviewer #1: (No Response)

Reviewer #2: Thank you for the authors' response. I believe that all the reviewers' questions have been answered and the text has substantially improved. Regarding the content of the article, I consider it ready for publication.

7. PLOS authors have the option to publish the peer review history of their article (what does this mean?). If published, this will include your full peer review and any attached files.

Reviewer #1: No

Reviewer #2: **Yes: **Rodrigo Gomes da Rosa

---

## [Editor Report · Acceptance letter]

11 Jan 2021

PONE-D-20-29049R1 

Running prevalence in Portugal: Socio-demographic, behavioral and psychosocial characteristics 

Dear Dr. Pereira:

I'm pleased to inform you that your manuscript has been deemed suitable for publication in PLOS ONE. Congratulations! Your manuscript is now with our production department. 

Kind regards, 

on behalf of

Dr. Daniel Boullosa 

Academic Editor

PLOS ONE